



# OCTOPUS Database v.2

Alexandru T. Codilean[1,*], Henry Munack[1,*], Wanchese M. Saktura[1], Tim J. Cohen[1], Zenobia Jacobs[1], Sean Ulm[2], Paul P. Hesse[3], Jakob Heyman[4], Katharina J. Peters[5,6], Alan N. Williams[7,8], Rosaria B. K. Saktura[1], Xue Rui[1], Kai Chishiro-Dennelly[9], and Adhish Panta[9]

[1]School of Earth, Atmospheric and Life Sciences, and ARC Centre of Excellence for Australian Biodiversity and Heritage, University of Wollongong, Wollongong, NSW 2522, Australia
[2]College of Arts, Society and Education, and ARC Centre of Excellence for Australian Biodiversity and Heritage, James Cook University, Cairns, QLD 4870, Australia
[3]School of Natural Sciences, Macquarie University, Sydney, NSW 2109, Australia
[4]Department of Earth Sciences, University of Gothenburg, Gothenburg 41320, Sweden
[5]Global Ecology, College of Science and Engineering, and ARC Centre of Excellence for Australian Biodiversity and Heritage, Flinders University, Adelaide, SA 5001, Australia
[6]Department of Anthropology, University of Zürich, Zürich 8006, Switzerland
[7]School of Biological, Earth and Environmental Sciences, and ARC Centre of Excellence for Australian Biodiversity and Heritage, University of New South Wales, Sydney, NSW 2052, Australia
[8]EMM Consulting Pty Ltd, St Leonards, NSW 2065, Australia
[9]Kasna, Sydney, NSW 2000, Australia
[*]these authors contributed equally to this work

**Correspondence:** Alexandru T. Codilean (codilean@uow.edu.au)

**Abstract.** OCTOPUS v.2 is an Open Geospatial Consortium (OGC) compliant web-enabled database that allows users to visualise, query, and download cosmogenic radionuclide, luminescence, and radiocarbon ages and denudation rates associated with erosional landscapes, Quaternary depositional landforms and archaeological records, along with ancillary geospatial (vector and raster) data layers. The database follows the FAIR (Findable, Accessible, Interoperable, Reusable) data principles and is

based on open-source software deployed on Google Cloud Platform. Data stored in the database can be visualised, queried, and downloaded via a custom-built web interface and via desktop GIS applications that support OGC data access protocols. OCTOPUS v.2 hosts five major data collections. CRN Denudation and ExpAge consist of published cosmogenic $^{10}$Be and $^{26}$Al measurements in modern fluvial sediment and glacial samples, respectively. Both collections have a global extent and in addition to geospatial vector layers, the former also includes raster layers, including digital elevation model, gradient raster, flow-

direction and flow-accumulation rasters, atmospheric pressure raster, and CRN production scaling and topographic shielding factor rasters. SahulSed consists of published optically stimulated luminescence (OSL) and thermoluminescence (TL) ages for fluvial, aeolian, and lacustrine sedimentary records across the Australian mainland and Tasmania. SahulArch consists of published OSL, TL, and radiocarbon ages for archaeological records and FosSahul consists of published late Quaternary records of direct and indirect non-human vertebrate (mega)fauna fossil ages that have been systematically quality rated. Supporting data

are comprehensive and include bibliographic, contextual, and sample preparation and measurement related information. In the case of cosmogenic radionuclide data, OCTOPUS also includes all necessary information and input files for the recalculation of denudation rates using the open-source program CAIRN. OCTOPUS v.2 and its associated data curation framework allow



the harnessing of valuable legacy data that would otherwise be lost to the research community. The database can be accessed at https://octopusdata.org (last access: 30 January 2022). The individual data collections can also be accessed via their respective

digital object identifiers (DOIs).

## 1   Introduction

Cosmogenic radionuclide (CRN) exposure dating (Granger et al., 2013; Schaefer et al., in press, 2022), luminescence dating (Rhodes, 2011; Murray et al., 2021), and radiocarbon dating (Hajdas et al., 2021) are geochronological techniques that are the most widely applicable to the recent geological past. All three of the techniques allow determination of the deposition age of

sediments and associated materials; and cosmogenic radionuclides can also be used to quantify the rate at which landforms or landscapes are lowered by physical and chemical erosion processes. The three techniques have made important contributions to the reconstruction of past environments (Roberts et al., 2001; Singhvi and Porat, 2008; Balco, 2019; Hocknull et al., 2020) and CRN and luminescence dating have revolutionised the field of quantitative geomorphology (Granger and Schaller, 2014; Dixon and Riebe, 2014; Guralnik et al., 2015; King et al., 2016). Radiocarbon and luminescence dating, and to some

extent also CRN exposure dating have also made substantial contributions to archaeology (Akçar et al., 2008; Renfrew, 2011; Roberts et al., 2015), including to the debates on the timing of human evolution and migration (Granger et al., 2015; Clarkson et al., 2017; Jacobs et al., 2019; Zilhão et al., 2020; Crabtree et al., 2021). Like most geochronological techniques, the three dating techniques require specialised training, laboratories and equipment, and involve lengthy and costly sample preparation procedures. As a result, studies relying on CRN, luminescence, or radiocarbon techniques will often produce relatively small

datasets ($n < 100$) that address very specific research questions that focus on relatively small study areas. For example, of the 285 publications reporting CRN-derived denudation rates over the past 25 years only 18 include datasets of $n \geq 50$ $^{10}$Be measurements with the median number of data points per publication staying constant over this period at a value of $\sim 15$. Furthermore, the lack of formal reporting standards (Schaefer et al., in press, 2022; Murray et al., 2021; Hajdas et al., 2021) coupled with the *disconnect* that exists in some cases between the researchers collecting the samples and interpreting the ages

and/or rates, and those preparing the samples and undertaking the measurements, means that the techniques often produce datasets that are unmanaged and that (i) may become *forgotten* once the study has been completed and results are published, and (ii) may not include sufficient levels of supporting information for the quality of the raw data to be easily determined or for the raw data to be reusable with confidence — for example in instances where data needs to be recalculated due to updated measurement standards and/or data reduction protocols. The above limitations mean that carefully curated compilations of

CRN, luminescence, and radiocarbon data are necessary to allow for larger-scale synoptic studies and instances where quality rating of ages / denudation rates is desirable, and are critical to ensuring the longevity and value of often irreplaceable legacy data.

In 2018, we published the OCTOPUS database (Codilean et al., 2018), consisting of a global compilation of cosmogenic $^{10}$Be and $^{26}$Al measurements from modern fluvial sediment and a compilation of optically stimulated luminescence (OSL)

and thermoluminescence (TL) measurements from fluvial sediment archives from Australia. The database was hosted at the





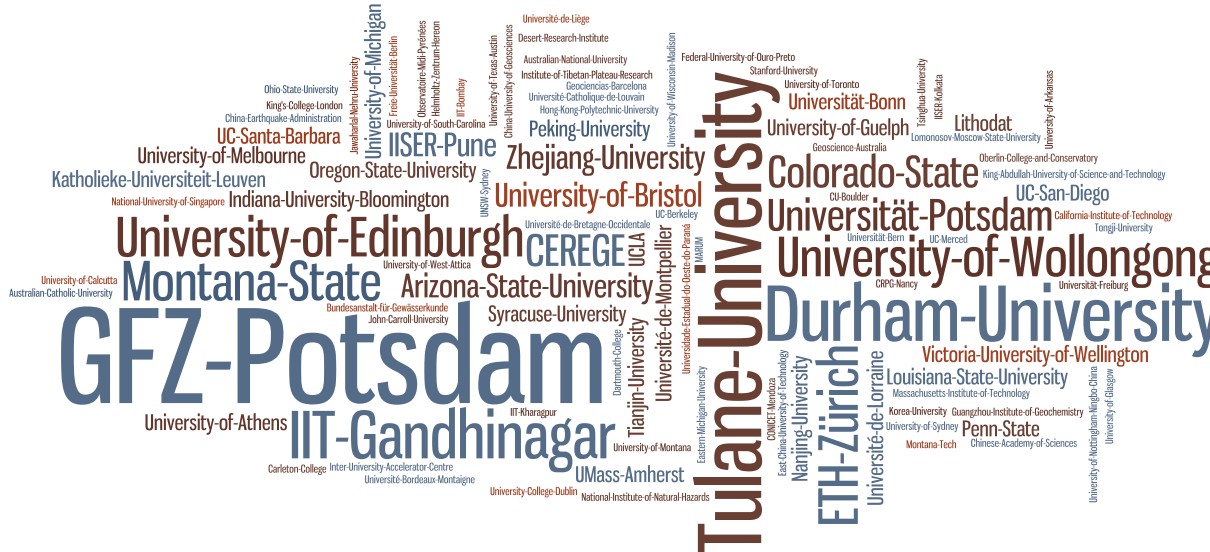

**Figure 1.** List of institutions whose members (researchers and students) requested data from OCTOPUS up to end of year 2021. Size of words is proportional to the number of unique individuals requesting data from each institution rather than the number of individual requests.

University of Wollongong and served to the research community via an Open Geospatial Consortium (OGC)-compliant web service (https://www.opengeospatial.org, last access: 30 January 2022). Since its launch, the OCTOPUS database has become an important resource to the global geomorphology community (Figure 1), logging over 900 data requests, mainly for CRN data to be used for both research ($\sim 80\%$ of requests) and classroom teaching ($\sim 20\%$ of requests), and, as intended, it has enabled

several regional- to global-scale synoptic studies (Dongen et al., 2019; Godard et al., 2019; Sternai et al., 2019; Delunel et al., 2020; Fülöp et al., 2020; Chen et al., 2021; Codilean et al., 2021a; Godard and Tucker, 2021). Here we describe the upgraded and updated version of the database — OCTOPUS v.2. The application part of the database was extensively rewritten, and is now running on Google Cloud Platform (https://cloud.google.com, last access: 30 January 2022). The data are stored in a relational database and the data collections have been extended to include a global collection of CRN exposure ages on

glacial landforms; an Australian collection of OSL and TL ages from aeolian and lacustrine sedimentary archives; OSL, TL, and radiocarbon ages from Sahul (Australia, New Guinea, and the Aru Islands joined by lower seal levels) archaeological records; and a collection of late Quaternary records of direct and indirect non-human vertebrate fauna fossil ages from Sahul. Supporting data are comprehensive and include bibliographic, contextual, and sample preparation and measurement related information. In the case of fluvial sediment CRN data, the database also includes all necessary information and input files for

the recalculation of denudation rates using CAIRN, an open-source program for calculating basin-wide denudation rates from $^{10}$Be and $^{26}$Al data (Mudd et al., 2016). Further, all CRN data have been recalculated and harmonised using the same program. OCTOPUS v.2 can be accessed at https://octopusdata.org (last access: 30 January 2022).





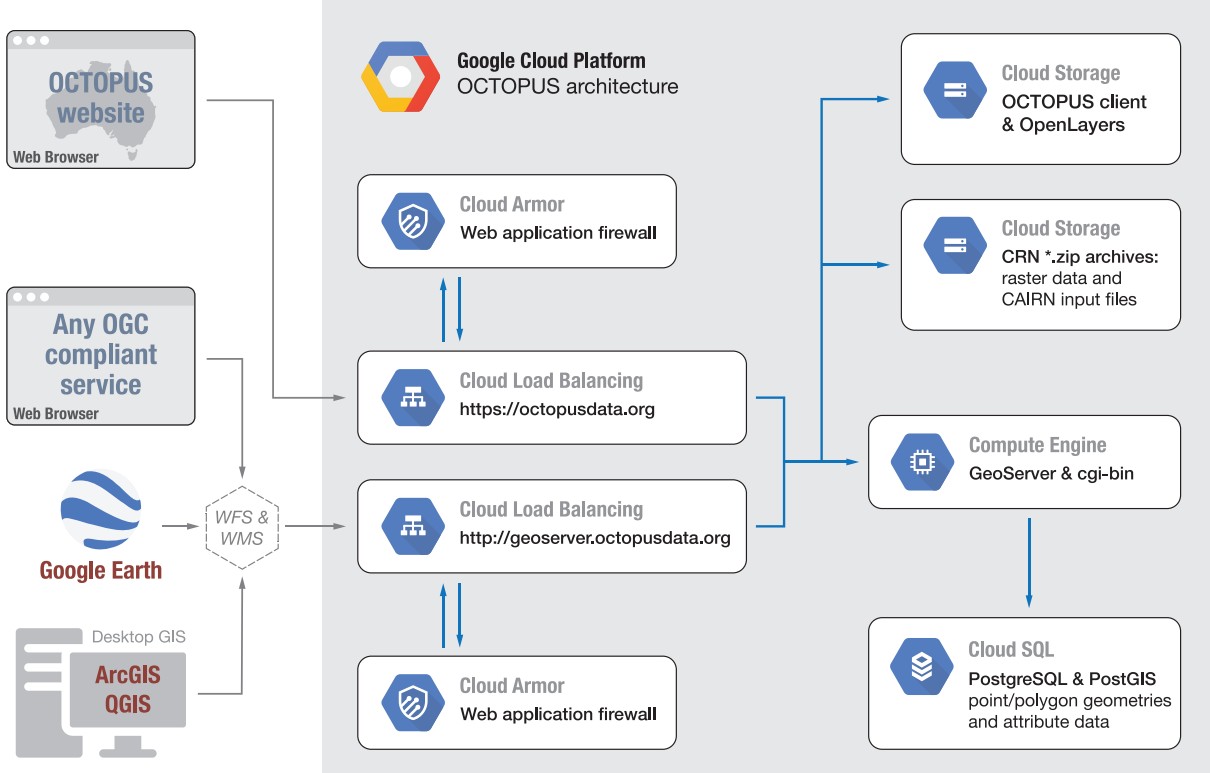

**Figure 2.** Schematic of the OCTOPUS v.2 Google Cloud Platform (GCP) setup. See text for more details.

## 2 System architecture

The software architecture behind OCTOPUS v.2 is illustrated in Figure 2. The software and data are deployed on Google Cloud
Platform (GCP) and follow a modular setup aimed at optimal leveraging of cloud services available within GCP. Although
migration of the OCTOPUS platform to a cloud hosted infrastructure such as GCP adds complexity to the system architecture,
Google Cloud offers extensive infrastructure and software solutions which are constantly updated with the latest technologies
and architectures. This constant evolution ensures that any future work and redesigns of the OCTOPUS platform have access to
best-in-class solutions. Further, the OCTOPUS platform is completely reproducible with access to a GCP environment, as the
source code contains the entire project and required documentation including infrastructure definitions, application definitions,
and deployment steps.

Most components of OCTOPUS v.2 run natively on GCP apart from GeoServer and Tomcat that are deployed within a
Google Compute Engine using a single bespoke Docker container (https://www.docker.com, last access: 30 January 2022).
Tabular data and the point and polygon geometries associated with each observation (see below) are stored in a PostgreSQL /
PostGIS (https://postgis.net, last access: 30 January 2022) relational database running in Cloud SQL. The latter is a SaaS (Soft-
ware as a Service) meaning that installation, setup, and running activities of the database are automatically managed by GCP,

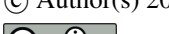



decreasing maintenance overhead and providing a monthly uptime/availability of $99.95\%$. Raster data and all CAIRN input and output files are stored separately within a Cloud Storage bucket in *.zip archives. Unlike with the first version of OCTOPUS (Codilean et al., 2018), the zip archives no longer include the tabular and vector data that is hosted in the PostgreSQL / PostGIS

relational database. Thus, we avoid duplication and make future maintenance of the data more straightforward. The relational database is linked to a GeoServer instance (Figure 2). GeoServer (http://geoserver.org, last access: 30 January 2022) implements a range of OCG data-sharing standards, including the widely used Web Feature Service (WFS) and the Web Map Service (WMS) standards that allow, in addition to connections from a web browser, direct connections to the database from a variety of desktop GIS applications, including ArcGIS and QGIS (via WFS; see below) and Google Earth (via WMS). GeoServer exports

data to various formats, including GML, JSON, Google Earth KLM and KMZ, and ESRI shapefile. GeoServer (together with Tomcat) is hosted in a Google Compute Engine, an IaaS (Infrastructure as a Service) that allows for a virtualized environment to be run on Google hardware. Geoserver and Tomcat currently exist as a single bespoke Docker container due to limitations of the deployed Geoserver and Tomcat versions that cannot run in separate runtimes. More recent Geoserver and Tomcat versions, however, exist as standard Docker containers that can be run independently aligned with a microservice architecture. Utilising

these dockerised versions would permit for the applications to be run on managed serverless platforms such as Google Cloud Run, allowing modular horizontal scaling. Further, Tomcat's CGI-Bin that provides functionality to the OCTOPUS frontend such as downloading files and retrieving study bounding boxes, could also be separated into independent resources thar run on Google Cloud Functions that also allow for near infinite horizontal scalability to meet any fluctuations in traffic volume. Next, the OCTOPUS web frontend is deployed in a Cloud Storage bucket and uses the OpenLayers (https://openlayers.org, last

access: 30 January 2022) JavaScript library to display the geospatial data served by the GeoServer instance in a web browser (Figure 2). Finally Cloud Load Balancing is used to distribute traffic and to separate connections to the web interface from those directed to GeoServer directly via WFS/WMS from third party applications.

## 3   Semantic data model

Unlike the prior version of the OCTOPUS database that stored data in a series of flat data tables (Codilean et al., 2018),
OCTOPUS v.2 builds on a fully relational PostgreSQL database that, using PostGIS spatial extensions, organises data following a two-pronged conceptual model (Figure 3). First, data are organised hierarchically going from a broader defined agglomeration of *sites* sharing common properties (= *metasite*) down to *observations*, namely, the actual $^{10}$Be, $^{26}$Al, OSL, TL, or radiocarbon age or rate data. Second, data are also organised thematically into (i) *local* data, spatial features, and parent tables — all of these serving a single data collection; (ii) *thematic* parent tables serving multiple data collections that are thematically linked

(e.g., are based on the same method, etc.); and (iii) *global* parent tables that serve all data collections (Figure 3).

In terms of hierarchy, the OCTOPUS v.2 data model includes four levels: metasite, site, sample, and observation. Whilst sites, samples, and observations apply to all data collections, metasites do not apply to the CRN Denudation and Sahul Sedimentary Archives (SahulSed) collections. A site, the hierarchical level subordinate to metasite, is a geographic point entity from which $n \geq 1$ samples have been collected. Therefore, sites without associated samples do not exist. A *site* is predominantly defined by



**Figure 3.** Representation of the OCTOPUS v.2 semantic data model. The full database schema along with HTML documentation is available in Munack and Codilean (2022). Inset refers to the 'Glen Lossie' metasite. See text for more details.



geographic attributes, including georeferencing information (e.g., country, region, island, river basin, coordinates, elevation) and other addressing / identification information (e.g., site name, alternative name and type of site). All site description data are stored in one global table. *Samples* represent the material — such as for example, shell, bone, rock fragment, river sand – that was collected and used for the age / denudation rate determination. Therefore, samples are (or were) a tangible entity. In OCTOPUS v.2, samples are described by sets of data-collection-specific attributes meaning that each data collection will have its dedicated sample table that links records to sites via unique site identifiers. Typical sample table attributes deal with physical sample properties (e.g., grain size, material dated, sample thickness, or density) and their very local depositional contexts (e.g., facies, shielding, depth below surface, excavation square or unit). Finally, *observations* — i.e., the actual age / denudation rate data — are stored in dedicated method specific tables that include fields aimed at capturing any meaningful auxiliary data that helps evaluate the quality of the age / denudation rate, and where this is necessary, further allow for the latter to be recalculated / reproduced.

We illustrate how the above hierarchical semantic data model is implemented in OCTOPUS v.2, using the example of a South Australian shell midden cluster (Wilson et al., 2012) (Figure 3, inset). A cluster of shell middens that share contextual similarities form a *metasite* — 'Glen Lossie' — that has a footprint that may be defined by a bounding box. Individual middens belonging to Glen Lossie are considered *sites* (point geometry) and have unique OCTOPUS site identifiers assigned (Figure 3, inset). Shell fragments are *samples* from those midden sites. In the Glen Lossie case, a repeat measurement was done on a shell fragment with the original ID 'GLM3-ss14'. As a result, OCTOPUS considers 'GLM3-ss14' and 'GLM3-ss14(r)' as a single sample with, however, two belonging observations, i.e., two separate radiocarbon ages (Obs. IDs ARCH0171C14001 and ARCH0171C14002, respectively; Figure 3, inset).

To serve the data collected in the OCTOPUS database as geospatial layers via an interactive map interface and to allow for manipulations of the data via the WFS protocol, each data sub-collection is served to GeoServer as a flat data table. The deployed version of GeoServer does not accept dynamically generated PostgreSQL virtual tables (knows as *views*) and so generating static flat data tables was required to serve the purpose of a "view". Newer versions of GeoServer, however, accept materialised views and an upgrade would present a possible improvement in the database by eliminating the need to store duplicate data. When downloading data from OCTOPUS, users are presented with point or polygon geospatial data files with associated attribute tables. Codilean et al. (2021b) provide field descriptions for each flat data table ($n = 19$). Direct connections to the PostgreSQL / PostGIS database are possible upon request. Munack and Codilean (2022) provide a complete documentation of the relational database, including a detailed database model diagram and searchable HTML documentation generated using SchemaSpy (https://schemaspy.org; last access: 30 January 2022).

## 4 CRN data recalculation

CRN-based exposure ages and denudation rates require periodic recalculation as measurement standards and calculation protocols are regularly revised and updated (Phillips et al., 2016; Schaefer et al., in press, 2022). Further, recalculating exposure ages and denudation rates is also necessary when comparing results produced by different accelerator mass spectrometry (AMS)

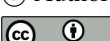



facilities that happen to normalise results to different AMS standards (Balco et al., 2008). To this end, the published $^{10}$Be and $^{26}$Al data included in the OCTOPUS database have been recalculated so that nuclide concentrations and denudation rates

are internally consistent and comparable. For completeness, the database also includes $^{10}$Be and $^{26}$Al concentrations and denudation rates as published. $^{10}$Be and $^{26}$Al concentrations ($\mathrm{atoms\,g^{-1}}$) were renormalised to the Nishiizumi 2007 $^{10}$Be AMS standard (Nishiizumi et al., 2007) and to the Nishiizumi 2004 $^{26}$Al AMS standard (Nishiizumi, 2004), respectively. Basin wide denudation rates were recalculated with the open-source program CAIRN (Mudd et al., 2016) with the following parameter settings: (i) nuclide production from neutrons and muons was calculated with the approximation of Braucher et al. (2011) using

a sea-level and high-latitude total production rate of $4.3\ \mathrm{atoms\,g^{-1}\,yr^{-1}}$ for $^{10}$Be and of $31.1\ \mathrm{atoms\,g^{-1}\,yr^{-1}}$ for $^{26}$Al; (ii) latitude and altitude scaling factors were calculated using the time-independent Lal/Stone scaling scheme (Stone, 2000) with atmospheric pressure calculated via interpolation from the NCEP2 reanalysis data (Compo et al., 2011); and (iii) topographic shielding was calculated from the same DEM using the method of Codilean (2006). Although several CRN denudation rate calculators are available (Balco et al., 2008; Vermeesch, 2007; Charreau et al., 2019), we prefer the CAIRN program for sev-

eral reasons. First, CAIRN is open-source and packaged in freely available software that runs on all commonly used operating systems. Second, CAIRN is automated and designed to allow for reproducibility of results. Users can simply publish a digital elevation model of their study area, CRN data files, and CAIRN input files, and denudation rates should be reproducible. Third, the open-source framework means that the code can be modified to include updated methods for production rates and scaling factors. Future users can thus recalculate denudation rates using updated versions of the code and the raster data and CAIRN

input files that are provided via OCTOPUS v.2 (see below), meaning that the CRN data will remain reproducible / reusable into the future.

## 5 The OCTOPUS v.2 data collections

The data in OCTOPUS v.2 are organised in three major collections: (i) *CRN Denudation* — a global collection of publicly available cosmogenic $^{10}$Be and $^{26}$Al measurements in modern fluvial sediment and respective basin-averaged denudation

rates, (ii) *Sahul Sedimentary Archives (SahulSed)* — a collection of publicly available OSL and TL ages for fluvial, aeolian, and lacustrine sedimentary records from Australia, and (iii) *Sahul Archaeology (SahulArch)* — a collection of publicly available radiocarbon, OSL, and TL ages for archaeological records from Sahul. Each of the above collections is further organised in several sub-collections based on geographic area, method, or sedimentary archive type (Table 1). In addition to the above collections — that we refer to as the OCTOPUS v.2 *core* data collections — the database also includes two *partner* collections,

namely, (i) *FosSahul* — a collection of publicly available quality ranked ages for the Late Quaternary non-human vertebrate fauna fossil records of Sahul, and (ii) *ExpAge* — a global collection of publicly available cosmogenic $^{10}$Be and $^{26}$Al measurements in glacial samples and respective recalculated exposure ages. The two partner collections have been fully integrated into the OCTOPUS v.2 relational database, however, currently these are not maintained nor officially supported by the OCTOPUS project and so versions that are more up to date (albeit less rich in auxiliary data) may exist elsewhere.





**Table 1.** Summary of OCTOPUS v.2 core data collections and sub-collections.

| Collection | DOI | Reference | Publication year range | % complete[1] | No. of observations | No. of data fields[2] | % 'no data'[3] |
|---|---|---|---|---|---|---|---|
| ***CRN Denudation*** | | | | | | | |
| Global | doi.org/10.25900/g76f-0h45 | Codilean and Munack (2021a) | 1996 – 2020 | 75% (77%) | 4,152 | 91 (45) | 3% (79%)[4] |
| Australia | doi.org/10.25900/mpr9-yn15 | Codilean and Munack (2021b) | 1998 – 2021 | >99% | 273 | 91 (45) | 0.3% (25%)[4] |
| | | | | | | | |
| ***Sahul Sedimentary Archives (SahulSed)*** | | | | | | | |
| Fluvial OSL | doi.org/10.25900/p5ye-rn35 | Cohen et al. (2021f) | 1997 – 2020 | >99% | 1,212 | 152 (76) | 56% |
| Fluvial TL | doi.org/10.25900/2a76-vw55 | Cohen et al. (2021e) | 1986 – 2020 | >99% | 564 | 150 (74) | 67% |
| Aeolian OSL | doi.org/10.25900/5jcw-tn50 | Cohen et al. (2021b) | 1993 – 2019 | >99% | 772 | 152 (76) | 56% |
| Aeolian TL | doi.org/10.25900/a2k9-kj43 | Cohen et al. (2021a) | 1987 – 2018 | >99% | 361 | 150 (74) | 66% |
| Lacustrine OSL | doi.org/10.25900/6hmv-zz61 | Cohen et al. (2021d) | 1997 – 2020 | >99% | 474 | 152 (76) | 48% |
| Lacustrine TL | doi.org/10.25900/32de-mj32 | Cohen et al. (2021c) | 1991 – 2015 | >99% | 41 | 150 (74) | 66% |
| | | | | | | | |
| ***Sahul Archaeology (SahulArch)*** | | | | | | | |
| Radiocarbon | doi.org/10.25900/2mb4-rr36 | Saktura et al. (2021c) | 1961 – 2020 | 51% (52%) | 5,039 | 120 (53) | 80% |
| OSL | doi.org/10.25900/ypr0-j711 | Saktura et al. (2021a) | 1990 – 2020 | 63% (41%) | 347 | 143 (76) | 46% |
| TL | doi.org/10.25900/xq40-t003 | Saktura et al. (2021b) | 1972 – 2019 | 63% (56%) | 127 | 141 (74) | 55% |

[1] Percent of studies and observations (in brackets) captured compared to total number of studies and observations published up to end of year 2021. For example, the CRN Denudation Global collection captures 75% of publications and 77% of data points that were available in the literature at the end of 2021.

[2] Values in parentheses represent number of fields containing information specific to the dating method (i.e., sample preparation and measurement related information), whereas values outside of parentheses represent total number of fields in each data table. The latter includes bibliographic details (i.e., details of publications from where data was collected) and contextual information related to samples and sample sites.

[3] Percent of records that are missing data due to information not reported in source. Refers only to fields containing information specific to the dating method.

[4] Values in parentheses indicate percent of "no data" entries for both $^{10}Be$ and $^{26}Al$, whereas values outside of parentheses consider only $^{10}Be$. The relatively large values in parentheses are due to most studies only including $^{10}Be$ measurements and $^{26}Al$ data fields for those studies being designated as "no data".

## 5.1 CRN Denudation

The CRN Denudation collection is composed of four sub-collections, two of which are officially supported as part of the OCTOPUS project and have been minted DOIs, namely, CRN Denudation *Global* and CRN Denudation *Australia*. The other two sub-collections — namely, CRN Denudation *Large Basins* and CRN Denudation *UOW (in preparation)* — are included only for completeness and have been described in more detail in Codilean et al. (2018). CRN Denudation Global and Australia consist of cosmogenic $^{10}Be$ (and where available, also cosmogenic $^{26}Al$) basin-wide denudation rates published in the peer-reviewed literature up to the year 2021. The two sub-collections include a total of $4,425$ $^{10}Be$ data points (Table 1 and Figure 4) with $1,083$ of those added as part of the update to version 2 ($n = 878$ for CRN Denudation Global, and $n = 205$ for CRN Denudation Australia). The Australian sub-collection captures every known study published up to 2021, whereas the global sub-collection includes about $77\%$ of all published $^{10}Be$ data ($75\%$ of publications). In terms of $^{26}Al$ data, the two sub-collections





include $403$ data points (Figure 4), of which $164$ were added as part of the update to version 2 ($n = 18$ for CRN Denudation Global, and $n = 146$ for CRN Denudation Australia). As with OCTOPUS v.1, in addition to the vector and attribute data that is stored in the PostgreSQL / PostGIS database, the CRN Denudation collection also includes seven raster data layers and a series of text files representing CAIRN configuration and input / output files (Table 2). These additional files are organised in "studies" — each study represents one publication — and are saved as separated zip archives with names keyed to the unique identifier assigned to each study (Codilean et al., 2021b). For consistency across the CRN Denudation collections, published $^{10}$Be and $^{26}$Al concentrations ($\mathrm{atoms\,g^{-1}}$) were renormalised to the same AMS standards and basin-wide denudation rates ($\mathrm{mm\,kyr^{-1}}$) were recalculated with the open-source program CAIRN (Mudd et al., 2016).

In terms of geographical extent, the updated CRN Denudation collection is similar to the previous version (Codilean et al., 2018): the majority of data are from Northern Hemisphere drainage basins, clustering around distinct, mostly tectonically active regions, such as the Pacific coast of the United States, the Appalachians, the European Alps, and the Tibet-Himalaya region (Figure 4A). There is also good coverage of the South American Cordillera. Large gaps in data from low-gradient and tectonically passive regions remain, although, OCTOPUS v.2 includes substantially more data from Australia and southern Africa. Despite the geographical bias, the CRN Denudation collection includes basins with a wide range of slope gradients, elevations, and basin areas (Figure 4B).

## 5.2 Sahul Sedimentary Archives (SahulSed)

SahulSed replaces the OSL & TL Australia data collection (DOI: 10.4225/48/5a836db1ac9b6) that was part of OCTOPUS v.1 (Codilean et al., 2018). The latter consisted of OSL and TL measurements in fluvial sediment samples from stratigraphic sections and sediment cores from across the Australian mainland and Tasmania, and included data published in the peer-reviewed literature up to 2017 and previously unpublished data compiled from technical reports and various Honours, MSc, and PhD theses. The OSL & TL Australia collection also included four raster layers, namely, a hydrologically corrected DEM, flow direction and flow accumulation rasters, and a slope gradient raster that were organised in zip archives (Codilean et al., 2018). Given the size of the various raster layers and the lack of demand for them from the user community (demonstrated by the lack of download requests), the raster layers have been dropped from SahulSed. The new data collection brings the fluvial data up to date in a new sub-collection — *SahulSed FLV* —, incorporating OSL and TL ages published since 2017. Further, SahulSed has been expanded to also include OSL and TL ages from aeolian (*SahulSed AEN*) and lacustrine (*SahulSed LAC*) sedimentary records, respectively. The aeolian sub-collection builds off a pre-existing database of luminescence ages of Australian desert dunefields (Hesse, 2016; Lancaster et al., 2016). SahulSed consists of a total of $3,426$ observations (see Table 1), of which $\sim 71\%$ are OSL ($n = 2,458$) and the remainder TL ages ($n = 968$).

In terms of geographical distribution of samples, data for Western Australia is generally lacking except for a few data points in the north-western (SahulSed FLV and AEN) and south-western (SahulSed LAC) parts of the state (Figure 5A—C). No lacustrine samples are present in Tasmania and the number of fluvial samples is also negligible as compared to the rest of the FLV sub-collections (Figure 5A—C). However, Tasmanian aeolian samples account for $\sim 10\%$ of the SahulSed AEN population. About $10\%$ of aeolian OSL and as little as $\sim 5\%$ of aeolian TL samples stem from the eastern seaboard, i.e., areas

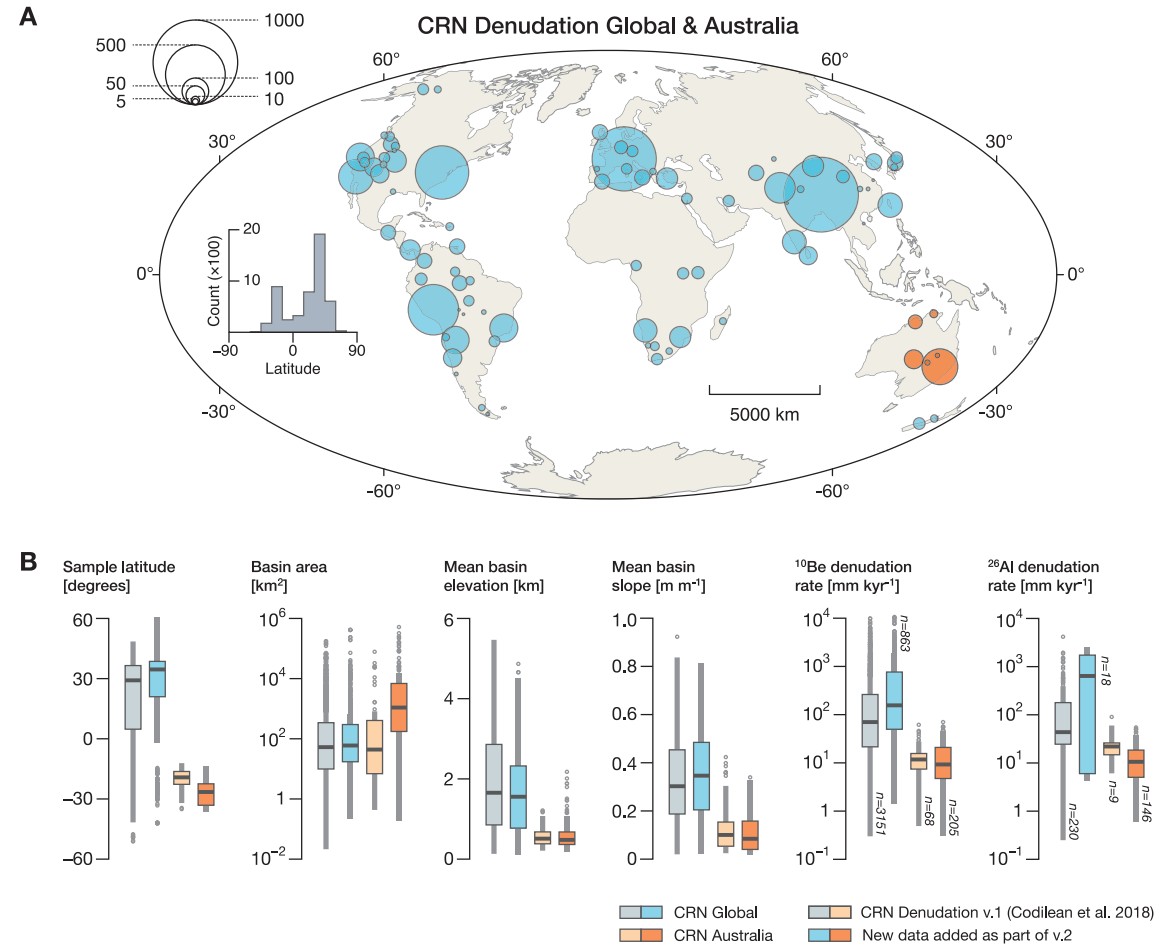

**Figure 4.** The CRN Denudation data collection. The size of circles corresponds to the number of observations in each 250 km radius cluster. Blue circles: CRN Global, orange circles: CRN Australia. Box plots are standard interquartile range (IQR) plots with outliers defined as data points $> 1.5 \times$ IQR. Box plots in blue hues: CRN Global sub-collection, box plots in orange hues: CRN Australia sub-collection; desaturated colours: data available as part of OCTOPUS v.1 (Codilean et al., 2018), saturated colours: data added as part of OCTOPUS v.2.




**Table 2.** Description of CRN data files available as part of *.zip packages.

| File name | Description | Reference |
|---|---|---|
| | ***CAIRN Input / Output ASCII files*[1]** | |
| s###_CRNData.csv | CAIRN input file containing sample names and locations as well as the measured $^{10}$Be (or $^{26}$Al) concentrations, uncertainties, and AMS standardisation. | Mudd et al. (2016) |
| s###_CRNRasters.csv | CAIRN input file containing path to DEM and topographic shielding raster. | Mudd et al. (2016) |
| s###_CRNResults.csv | CAIRN output file containing calculated $^{10}$Be (or $^{26}$Al) denudation rates and nuclide production scaling parameters. | Mudd et al. (2016); Balco et al. (2008); Vermeesch (2007) |
| s###_CRONUSInput.txt | Input file for the online calculators formerly known as the CRONUS-Earth on-line calculators. File produced by CAIRN. | Balco et al. (2008) |
| s###_CRNParam | CAIRN parameter file. | Mudd et al. (2016) |
| | ***ENVI BIL raster layers*[2]** | |
| s###_atmospres.* | Atmospheric pressure raster, showing local atmospheric pressure in hPa calculated based on the NCEP2 climate reanalysis data. | Compo et al. (2011) |
| s###_d8flowdir.* | Flow-direction raster calculated using the D8 flow-routing method. | Jenson and Domingue (1988) |
| s###_demhydro.* | Hydrologically corrected Shuttle Radar Topography Mission (SRTM) DEM with elevation values in metres. | Farr et al. (2007) |
| s###_flowacc.* | Flow-accumulation raster calculated from the D8 flow-direction raster. | Jenson and Domingue (1988) |
| s###_gradmkm.* | Slope gradient ($\mathrm{m\,km^{-1}}$) raster calculated from the SRTM DEM. | Horn (1981) |
| s###_prodscale.* | Cosmogenic nuclide production scaling raster. | Stone (2000) |
| s###_toposhield.* | Cosmogenic nuclide production topographic shielding raster. | Codilean (2006) |

[1] In cases where both $^{10}$Be and $^{26}$Al data is available, two sets of input and output files will exist. Multiple sets of input / output files will also exist when a larger study has been broken up in smaller chunks.

[2] Raster layers have a resolution of 90 m except for studies with very large basins, in which case resolution will be either 250 or 500 m. For most Australian studies, data layers have a resolution of 30 m. Slope gradient rasters have a resolution of 90 m in all cases and were calculated from a 90 m resolution SRTM DEM even if other layers in the same study have resolutions of 30, 250, or 500 m, respectively. Raster layers are projected to the WGS84 Universal Transverse Mercator (UTM) coordinate system, with UTM zones varying based on location and extent of each study.





draining to the east of the Great Dividing Range. The picture is similar for lacustrine OSL and TL, with only $\sim 10\%$ of OSL

samples and no TL samples originating from east of the Great Dividing Range. In contrast, however, $\sim 55\%$ of fluvial OSL and $\sim 20\%$ of fluvial TL samples have been derived from the eastern seaboard. The above geographical distribution reflects, to a large extent, the natural distribution of sedimentary facies across Australia and where research interests were / are focused — for example focused interest on river systems proximal to high-population density areas, where floods are a potential threat, such as the eastern seaboard, or where rivers are of great agricultural importance, such as the Murray Darling Basin. To this end,

the aeolian sub-collections mainly cover the grassland and desert zones (Figure 5B), namely, the southern parts of the Northern Territory (mainly in the Diamantina-Georgina Rivers drainage system) and South Australia (Diamantina-Georgina, Cooper Creek-Bulloo, Lake Eyre, and the lower Murray-Darling fluvial systems). Samples of the FLV sub-collections, however, show good coverage of large parts of the northern and eastern Australian coast (Figure 5A). Additionally, there is noticeable overlap of aeolian and fluvial sample distribution in the central parts of Australia, namely, again, the Diamantina-Georgina drainage

system, the Lake Eyre basin and the South Australia – Queensland borderlands of the Cooper Creek-Bulloo drainage system (Figure 5A—B). Several additional fluvial data clusters exist in the upper and lower parts of the Murray-Darling Basin in New South Wales (Figure 5A). SahulSed lacustrine data, compared to the aeolian and fluvial sub-collections are sparse, with a clustering peak around Lake Eyre, Lakes Blanche, Callabonna and Frome in South Australia, the Lake Mungo and Lake George areas in New South Wales, Lakes Lewis and Woods in the Northern Territory, and Lake Gregory in Western Australia

(Figure 5C).

Aeolian and lacustrine sample OSL and TL age distributions (Figure 5D), compared between OSL and TL sub-collections, are similar with overlapping IQRs. While the age distribution of the fluvial TL sub-collection cannot be distinguished from the aeolian and lacustrine data, fluvial OSL ages are distributed across a wider range and IQR, with a median value that is approximately one order of magnitude lower than for its TL counterpart. SahulSed OSL ages spread across five orders of

magnitude ranging from tens of years back to the middle Pleistocene (0.001 ka to 716 ka for FLV OSL and AEN OSL sub-collections, respectively). Extreme TL age values are comparable (0.007 ka to 740 ka for FLV TL and AEN TL sub-collections, respectively).

## 5.3 Sahul Archaeology (SahulArch)

The SahulArch collection is composed of three sub-collections, namely, *SahulArch Radiocarbon*, *SahulArch OSL*, and *Sahu-*

*lArch TL* (Figure 6). The sub-collections comprise radiocarbon, OSL, and TL ages for archaeological records from Sahul published both in the peer-reviewed and grey literature up to the year 2021. SahulArch is $\sim 50\%$ complete (Table 1), and to date, data entry has focussed on (i) Australia-wide coverage of all ages older than $30,000$ years, (ii) ages published since 2014 regardless of age, and (iii) the geographical areas of northern and south-eastern Australia. Data collection and entry will be progressively expanded to cover the whole of Sahul, building off the pre-2014 dataset presented in *AustArch* (Williams et al.,

2014). To protect the location of culturally sensitive archaeological sites, sample coordinates have been randomly obfuscated within a $25$ km radius using the $NRand$ point obfuscation algorithm (Wightman et al., 2011; Zurbarán et al., 2018). To this



**Figure 5.** The SahulSed data collection: distribution of sample sites and ages. The size of circles corresponds to the number of observations in each 50 km radius cluster. Box plots are standard interquartile range (IQR) plots with outliers defined as data points $> 1.5 \times$ IQR. Box plots only include observations with non-zero ages. Box plot colours are as follow: blue — fluvial; red — aeolian, yellow — lacustrine.





end, the SahulArch samples are stored as polygon rather than point features, with the non-obfuscated sample coordinates stored in the PostgreSQL relational database but hidden from users.

The three SahulArch sub-collections include a total of $5,513$ observations (Table 1, Figure 6), comprising $5,039$ radiocarbon, $347$ OSL, and $127$ TL ages from $1,120$ archaeological sites (*metasites* in OCTOPUS v.2 parlance). In the radiocarbon sub-collection, $50\%$ of metasites have between 1 and 4 observations (median $= 2$). Only a handful of metasites have more than 10 radiocarbon dates and only two metasites have more than 100 radiocarbon dates (Figure 6B). The few metasites with OSL and TL ages, in general have more observations on average, the IQR and median being $\sim 50$ to $20$, and $\sim 7$ observations, respectively. In terms of geographical extent, the SahulArch collection is unique in encompassing the entire continent of Sahul, comprising Australia, Tasmania, New Guinea and islands connected to mainland Australia and New Guinea at times of lower sea levels (Figure 6A). Given that Australia was joined to New Guinea for most of the human history of the continent, the addition of New Guinea will articulate archaeological knowledge of New Guinea into analyses and interpretations that have, until now, been largely based on Australian data. The current as well as future expansions of SahulArch will further facilitate modelling efforts aimed to better understand the history of human occupation of the Sahul landmass. The utility of these data collections has been illustrated recently by two studies looking at the first peopling of Sahul (Bradshaw et al., 2021; Crabtree et al., 2021) that rely on a precursor of SahulArch (Williams et al., 2014) to provide chronological data for numerical modelling of possible peopling pathways across the landmass. The SahulArch collection is currently biased towards ages published since 2014 and the geographical areas of the northern and south-eastern Australia where data entry has focussed to date. The IQR of radiocarbon ages is between 1 to 10 ka whereas that of OSL and TL ages is higher, between 10 to 45 and 10 to 60 ka, respectively (Figure 6C).

## 5.4 Partner data collections

In addition to the three core data collections (above), OCTOPUS v.2 also includes two partner data collections, namely *FosSahul* and *ExpAge*. The two partner collections are not formally supported by the OCTOPUS project meaning that the versions included in the OCTOPUS v.2 database are forks of the 'official' versions and so might lag in terms of version currency. In contrast, however, the OCTOPUS versions of these collections have been fully integrated in the relational database — sharing the same semantic data model as the core data collections — and include more auxiliary data than the official versions (Codilean et al., 2021b). In terms of version currency, the OCTOPUS v.2 implementation of FosSahul corresponds to FosSahul v.3 (13 April 2021) with the DOI: 10.6084/m9.figshare.8796944. The OCTOPUS v.2 implementation of ExpAge corresponds to *expage-202006* located at: https://expage.github.io (last access: 30 January 2022).

FosSahul is a collection of publicly available Late Quaternary non-human vertebrate fauna fossil ages from Sahul. The collection includes $11,858$ records. Ages are quality rated based on the dating protocols that were used and the association between the dated materials and the fossil remains. FosSahul and the methodology behind the data collection and quality rating is described elsewhere (Rodríguez-Rey et al., 2016; Peters et al., 2019, 2021) and readers are referred to these publications for further details. As with SahulArch, sample coordinates have been randomly obfuscated within a $25$ km radius (Wightman et al., 2011; Zurbarán et al., 2018) and samples are stored as polygon rather than point features.

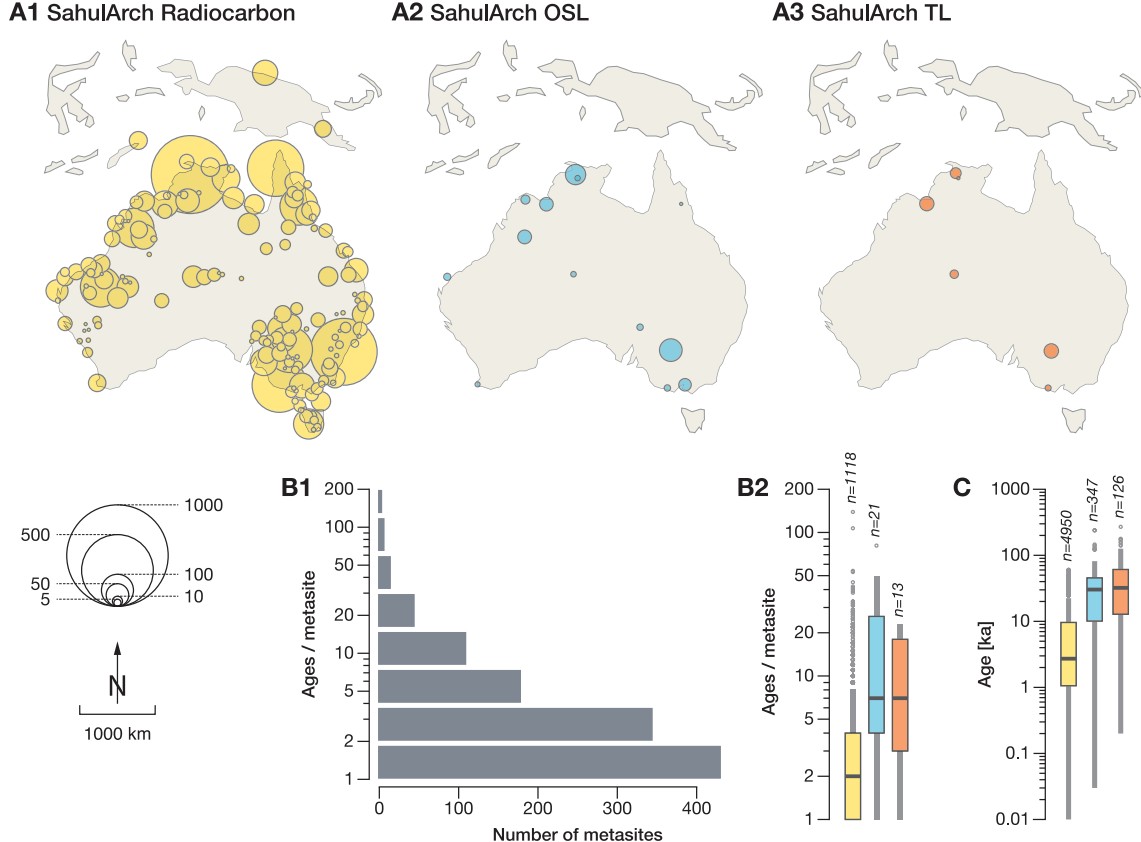

**Figure 6.** The SahulArch data collection. (A) Geographic distribution of sample sites. (B) Number of ages per metasite. (C) Distribution of ages. The size of circles corresponds to the number of observations in each 50 km radius cluster. Box plots are standard interquartile range (IQR) plots with outliers defined as data points $> 1.5 \times$ IQR. The frequency distribution plot in (B1) uses logarithmic bins for clarity. Box plots in (C) only include observations with non-zero ages. Box plot colours are as follow: yellow — radiocarbon; blue — OSL, red — TL. Note that the SahulArch collection is only $\sim 50\%$ complete (see text and Table 1).

ExpAge is a global collection of publicly available cosmogenic $^{10}$Be and $^{26}$Al measurements and associated sample data in glacial samples (e.g., erratic boulders, striated bedrock, etc.) and respective recalculated exposure ages. The OCTOPUS v.2 version of ExpAge includes $16{,}009$ observations (of which $2{,}229$ include both $^{10}$Be and $^{26}$Al data) published in the peer-reviewed literature between 1989 and 2020 from 766 primary publications. The ExpAge collection consists of $5{,}210$ metasites – each representing a distinct group of samples derived from a single location with one expected deglaciation age. The median number of observations per metasite is 2 with the IQR between 1 and 4. The maximum number of observations per metasite is 39 ($n = 1$) and of the $5{,}210$ metasites only $\sim 300$ include $\geq 10$ observations. The data collection's spatial sample distribution is determined by the global occurrence of past and recent ice bodies with key areas being the former North American ice sheets (19% of all observations), High Asia (17%), the former Eurasian ice sheet (14%), Antarctica (11%), Greenland (7%), the





Andes (7%), New Zealand (5%), the European Alps (5%), Patagonia (4%), Inner Asia (2%), and the Iberian peninsula (2%). Small data clusters (each of them accounting for less than one percent of the total sample population) exist where comparably small ice bodies left glacial landforms as for instance on the Australian mainland, in Tasmania, Morocco, or Japan. The most isolated sets of samples have been derived from the flanks of Costa Rica's Cerro Chirripó ($n = 9$) and the Rwenzori in Uganda ($n = 8$). More information about the ExpAge data collection, including the methodology used to recalculate exposure ages, is

available here: https://expage.github.io (last access: 30 January 2022).

## 6 Accessing data from OCTOPUS v.2

As with the previous version of OCTOPUS (Codilean et al., 2018), data can be accessed either via the bespoke web interface (Figure 7) or directly via the WFS capability running on GeoServer. OCTOPUS v.2 features a completely redesigned web interface with functionality added to filter the data using SQL queries and to export spatial data directly to various geospa-

tial vector data formats. Other notable additions include the ability to change the base map (four different base maps are available: three from MapTiler — https://www.maptiler.com, last access: 30 January 2022; and one from OpenStreetMap — https://www.openstreetmap.org, last access: 30 January 2022), and the clustering of sample locations with the ability of changing the cluster radius and toggling clusters on and off. The web interface consists of a map view and a collapsible side pane (Figure 7A) that contains five separate panels: (i) the *Layers* panel that displays a list of the available data layers and allows for

these to be toggled on and off; (ii) the *Filter* panel that allows users to query the data and limit what is displayed on the map to only results returned by the applied filters; (iii) the *Export Data* panel that allows users to download geospatial vector data layers; (iv) the *Download Collection* panel that allows users to download raster data and CAIRN input / output files associated with the CRN Denudation collection; and (v) the *Settings* panel where users can change the choice of base map and modify clustering settings. When designing the updated web interface, our aim was to offer users more functionality but at the same

time avoid replicating tools that are readily available within desktop GIS applications.

### 6.1 Accessing data using the web interface

The sequence of screenshots in Figure 7 illustrates a typical intended user interaction with the OCTOPUS v.2 web interface. First, the user displays the data collection(s) of interest and navigates to the desired geographical region (Figure 7, #1). Sample locations are displayed as clusters with the size of the circle being scaled with the number of features within the cluster (the

same information is also conveyed by the number written inside each circle). In the case of the CRN Denudation collections, basin outlines are displayed as translucent polygons. Clicking on a feature will display a dialog panel with key information about that feature, including sample identifiers, bibliographic details, information about the dating methods, and associated age /denudation rate, etc (Figure 7, B). Where a DOI is available, it is listed as a hyperlink and will connect to the publication from where the data was sourced. The pop-up dialog panel displays only a subset of the available attribute data and is meant

to provide the user with basic information about each point or polygon record. The dialog panel closes automatically once the user clicks anywhere outside of the panel in the map display window.



**Figure 7.** Annotated screenshots of the OCTOPUS v.2 web interface illustrating a typical use case. See text for more details.





To download data from OCTOPUS, the user may next access the Export Data panel (Figure 7, #3) and download one or more entire data collections, or access the Filter panel and prepare a subset of the data for download, first (Figure 7, #2). Filtering is automatically enabled for those data layers that are displayed on the map. SQL filters are entered as 'rule' blocks (Figure 7, D) and multiple blocks may be combined in groups. The latter allows for a virtually unlimited number of rules to be applied to a given layer as part of a virtually unlimited number of group configurations. For each set of rules and / or groups the user may also specify whether the $AND$ or the $OR$ SQL operators are to be used (Figure 7, C). The Filter panel provides the option for filters to be cleared both locally (affecting one data layer) or globally (i.e., clearing all filters) (Figures 7, F and H). Finally, users also have the option to copy the filter configuration as a WFS URL command that can be pasted in a browser window (Figure 7, E), or save the filter configuration to disk for re-loading in a different session or on a different machine (Figure 7, G).

To download vector geospatial data files, the user is asked to select, using a series of drop-down lists, the data collection of interest, the exported data format, and the intended use of the data. The latter information will be used for reporting purposes to funding organisations and when applying for further funding to support OCTOPUS. In terms of data formats, the following are available: Geography Markup Language (GML) versions 2 and 3, ESRI Shapefile, JavaScript Object Notation (JSON), and Google Earth KML and KMZ. When the KML and KMZ formats are selected, the exported data will be truncated to the displayed extent of the data. For the other data formats, the entire data layer is exported, unless filtering is enabled, in which case only those data entries are exported that are returned by the filter.

Those users who are interested in downloading the raster data and the CAIRN input and output files associated with the CRN Denudation data collections, may do so using the Download Collection panel. The procedure for downloading the raster / CAIRN files is the same as in the previous version of OCTOPUS (Codilean et al., 2018). When the Download Collection panel is active the cursor turns into a selection tool and the user drags a box around desired points and polygons to select. The user has the option to fine-tune the list of selected studies by toggling on or off each study from the list generated after the selection box is drawn. It is possible to select multiple studies from multiple collections at the same time (Figure 7, J). The user is also asked to enter a name, an email address, and an intended use of the data. A valid email address is required as links to the data are sent to the user via email immediately after the download button is pressed. There is no verification of who the data requestor is or where that person is from; however, none of the fields can be left empty and all entered information is stored in a log file. As stated above, the requested information is used for reporting purposes (see for example Figure 1) and by providing meaningful information when downloading the data, users will support efforts to secure future funding for updating and expansion of OCTOPUS.

## 6.2 Accessing data using the Web Feature Service (WFS) capability

A description of how to access data via WFS from OCTOPUS, including specific examples, is provided elsewhere by Codilean et al. (2018). Further, most of the core WFS functions have now been incorporated into the web interface as part of the Filter and Export Data panels. For the above reasons, below we only provide some basic information on connecting to OCTOPUS v.2 from a third-party application (e.g., QGIS or $R$). For a more comprehensive introduction to WFS and GeoServer, the reader is re-





ferred to Iacovella and Youngblood (2013) or to the GeoServer documentation web page accessible at http://docs.geoserver.org (last access: 30 January 2022).

With the migration of OCTOPUS to GCP, the platform is no longer hosted at the University of Wollongong and there is a new URL for accessing the data via WFS:

```
http://geoserver.octopusdata.org
        /geoserver/wfs
```

370   Probably the simplest way to access data from OCTOPUS via WFS is by using the WFS / OGC API in QGIS (https://qgis.org, last access: 30 January 2022): the only information required is the above URL. Those preferring the $R$ software environment (https://www.R-project.org, last access: 30 January 2022) may use the $ows4R$ package (Lovelace et al., 2020; Blondel, 2021) to connect to OCTOPUS via WFS. The following $R$ code snippet will establish a connection to the OCTOPUS database and fetch the list of available data layers:

```
library(ows4R)
OCTOPUSdata <-
  "http://geoserver.octopusdata.org
          /geoserver/wfs"
OCTOPUSdata_client <-
  WFSClient$new(
      OCTOPUSdata,
      serviceVersion = "2.0.0"
      )
OCTOPUSdata_client$
  getFeatureTypes(pretty = TRUE)
```

375   Next, the following code snippet may be used to send a WFS request to GeoServer and download some data. In the example below, the request will download all drainage basins belonging to the CRN Denudation Australia sub-collection:

```
library(sf)
library(httr)
url <- parse_url(OCTOPUSdata)
url$query <- list(
    service  = "wfs",
    version  = "2.0.0",
    request  = "GetFeature",
    typename = "be10-denude:crn_aus_basins",
    srsName  = "EPSG:900913"
    )
```



```
request <- build_url(url)
CRN_AUS_basins <- read_sf(request)
```

In the above example, `'typename'` tells GeoServer the name of the data layer to be served and `'srsName'` specifies the output coordinate system (in this case WGS 84 / Pseudo-Mercator). Additional instructions may be specified, such as for example, the desired file format (e.g., `outputformat=SHAPE-ZIP` for ESRI Shapefile) or if only a subset of the data is required, by using the CQL/ECQL query language (see GeoServer documentation).

## 7 Technical validation

As with the previous version of OCTOPUS, our aim was to compile and incorporate all data — both published and unpublished — that is publicly available. It is not our role to decide on the quality of the data that are already published and so we make no editorial decisions on what data to include or exclude. Further, we designed the database in a way that it captures sufficient auxiliary information for users to be able to make informed judgements regarding data quality, themselves. However, in some instances, where a publication did not provide sufficient information for the data files to be produced (e.g., insufficient information to be able to confidently locate and delineate drainage basins) and this information could not be obtained from elsewhere, those data were not included in OCTOPUS. Further, despite our best efforts and given the sheer volume of data present, some of the sub-collections that make up OCTOPUS v.2 are only partially complete (see Table 1) with the excluded data in the queue for the next version of the database.

When recalculating CRN-based denudation rates, for simplicity and consistency across the global compilation, we do not correct for lithological differences in quartz abundance, glacier cover, and snow shielding. Performing such corrections in a consistent manner on a global scale is impossible. However, as shown in Figure 8A, as with the basins included in the previous version of the OCTOPUS database (blue circles), the new basins added as part of version 2 (red circles) show a good agreement between the published and recalculated rates, with the IQR of the difference being between $2\%$ and $20\%$ (median $= 8\%$), values within the uncertainty on the calculated denudation rates. Further, since we provide all CAIRN input and configuration files, the above corrections can be readily applied by end users to individual studies if the discrepancy between published and recalculated denudation rates deems this necessary. For consistency with the previous version of the database, the $^{10}$Be and $^{26}$Al denudation rates included in the CRN Denudation collections were corrected for topographic shielding. A recent study by DiBiase (2018) suggested that topographic shielding corrections are inappropriate for calculating basin-wide denudation rates, in most settings, and are only required for steep catchments with non-uniform distribution of quartz and / or denudation rates. Notwithstanding the above, topographic shielding corrections are trivial: ignoring this correction in the basins with topographic shielding $> 90^{th}$ percentile ($n \approx 410$) results in a difference in calculated denudation rates between $3.5\%$ and $10\%$ (Figure 8B, green). In comparison the IQR of the uncertainty in the calculated denudation rates is larger, between $\sim 20\%$ and $\sim 25\%$ (Figure 8B, black). Therefore, although ignoring topographic shielding will produce higher denudation rates, these will be within the uncertainty of the values currently calculated in OCTOPUS v.2, and so a recalculation of the entire CRN Denudation collection without topographic shielding correction is not warranted.
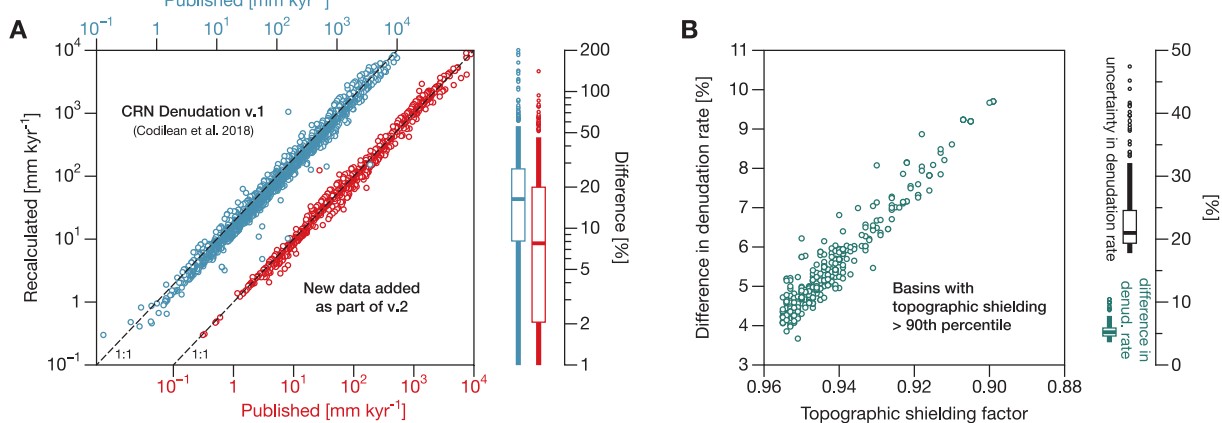

**Figure 8.** Technical validation of the CRN Denudation data collection. (A) Published versus recalculated [10]Be-based denudation rates. Blue circles represent [10]Be data from the first version of the OCTOPUS database (Codilean et al., 2018) and red circles represent the [10]Be data added as part of version 2. Box plots show % difference between published and recalculated [10]Be-based denudation rates (blue — data in v.1; red — data added as part of v.2). (B) Comparison of [10]Be-based denudation rates calculated with and without correcting for topographic shielding for basins from the OCTOPUS v.2 database with topographic shielding values above the $> 90^{th}$ percentile. Box plots show % difference in denudation rates calculated with and without correcting for topographic shielding — in green, compared to % uncertainty in calculated denudation rates — in black. Note how the difference in [10]Be denudation rate obtained by ignoring topographic shielding is smaller than the uncertainty in the denudation rate calculation.

Unlike CRN-based exposure ages and denudation rates, the OSL, TL, and radiocarbon data compiled as part of SahulSed and SahulArch do not require recalculation. For this reason, the two data collections are verbatim reproductions of what was present in the various sources that the data was compiled from. We do not quality-rate the published OSL, TL, or radiocarbon ages, however, to enable end users to undertake such quality rating if desired, we designed the database so that a wide range of method related data is captured. For example, for OSL and TL ages OCTOPUS v.2 includes 76 and 74 auxiliary data fields, respectively; for radiocarbon ages the number of auxiliary data fields is 53 (Table 1). Given the lack of formal data reporting standards, however, few publications report comprehensive supporting information. In the case of the radiocarbon sub-collection, the situation is made worse by the fact that a considerable proportion of the data was published in the grey literature or was published several decades ago and so in most cases supporting information is not provided. As such, about 80% of SahulArch radiocarbon auxiliary data fields are null (Table 1). For example, of the 5,039 compiled radiocarbon ages, 3,387 (67%) do not have information about the chemical pretreatment method used, and 2,778 (55%) do not have information about the $\delta^{13}$C fractionation value (Figure 9). Further, of those radiocarbon ages that were determined using AMS ($n = 1,704$), 72% do not have information about pMC / F[14]C values (Figure 9), meaning that the former are not reproducible and need to be taken at face value.



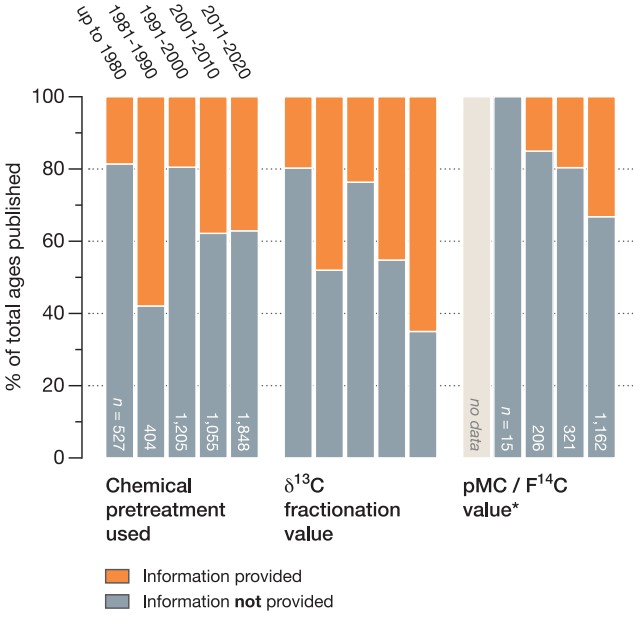

**Figure 9.** Radiocarbon-dating-related supporting information reported in archaeology studies published between 1961 and 2020 and included in the SahulArch Radiocarbon sub-collection. The statistics for pMC / F$^{14}$C value only include AMS radiocarbon ages. Note how most studies do not report information on chemical pretreatment method, $\delta^{13}$C fractionation value, or pMC / F$^{14}$C value.

## 8 Data availability

OCTOPUS v.2 can be accessed at https://octopusdata.org (last access: 30 January 2022). If connecting through WFS via third party apps (such as QGIS or $R$), users should use the following url: http://geoserver.octopusdata.org/geoserver/wfs (last ac-
425 cess: 30 January 2022). DOIs assigned to each sub-collection are listed in Table 1. Users should refer to the DOIs provided to ensure that they are accessing the current and supported version of the data. Supporting information, including field descriptions, detailed relational database model diagram, and searchable HTML documentation, is provided via Zenodo (Codilean et al., 2021b; Munack and Codilean, 2022). The OCTOPUS database is listed in the Registry of Research Data Repositories (https://www.re3data.org; last access: 30 January 2022), with listing DOI: 10.17616/R31NJN2E.

430 User contributions to OCTOPUS are welcome. Those wishing to contribute detrital CRN data should download a study and use that as the template for data structure, formats, and naming convention. As a minimum, a contribution should include point and polygon geometry files, and an attribute table with all records listed in the database documentation (Codilean et al., 2021b), except for those records that are output by CAIRN. Those wishing to contribute OSL, TL, and radiocarbon data should contact the authors to be provided with a data entry template. The data collections making up OCTOPUS v.2 have been assigned DOIs,
435 and therefore, adding new data needs to follow a versioning scheme, with each new version requiring new DOIs. Thus, data contributed by users will be incorporated in the next release of a given collection, rather than being added to the current one.



# 9 Conclusions

With the second version of the OCTOPUS database, we have extensively upgraded the software infrastructure and have updated and expanded the constituent data collections. The application part of the database was extensively rewritten, now running on Google Cloud Platform (https://cloud.google.com, last access: 30 January 2022). The data are stored in a relational database and the data collections have been extended to include a global collection of CRN exposure ages on glacial landforms; an Australian collection of OSL and TL ages from aeolian and lacustrine sedimentary archives; OSL, TL, and radiocarbon ages from Sahul archaeological records; and a collection of late Quaternary records of direct and indirect non-human vertebrate fauna fossil ages from Sahul. Since launch in 2018, the first version of the OCTOPUS database (Codilean et al., 2018) has become an important resource to the global geomorphology community (Figure 1), logging over 900 data requests. Although most data requests indicated intended use of data for research purposes, about 20% of requests were from undergraduate and graduate students as part of classroom teaching. We hope that with the expansion of SahulSed to include OSL and TL ages from aeolian and lacustrine sedimentary archives, in addition to the fluvial archives that were part of OCTOPUS v.1, and with the inclusion of SahulArch, OCTOPUS v.2 will become an equally important resource for the Australian Quaternary and archaeology research communities. The utility of these data collections has been illustrated recently by two high-profile studies looking at the first peopling of Sahul (Bradshaw et al., 2021; Crabtree et al., 2021) that rely on a precursor of SahulArch (Williams et al., 2014) to provide chronological constraints on numerical modelling and select most plausible modelled scenarios. Ultimately, it is our hope that OCTOPUS will continue to ensure that data are reusable beyond the scope of the project for which they were initially collected, and so continue to enable large-scale synoptic studies that would otherwise not be possible.

*Author contributions.* HM designed the data model with input from ATC, WMS, TJC, ZJ, and SU; ATC and HM compiled the CRN data; WMS, ZJ, and SU compiled the OSL, TL, and radiocarbon data that is part of SahulArch with data contributed by ANW; TJC, RBKS, and XR compiled the OSL and TL data that is part of SahulSed with data contributed by PPH; HM adapted the ExpAge and FosSahul collections to match the OCTOPUS data model with data compiled by JH (ExpAge) and KJP (FosSahul); ATC and HM designed the OCTOPUS v.2 platform and web interface; All authors contributed to the writing of the manuscript.

*Competing interests.* The authors declare that they have no conflict of interest.

*Acknowledgements.* We acknowledge financial support from the Australian Research Council Centre of Excellence for Australian Biodiversity and Heritage (ARC Grant CE170100015). We thank Rachel Wood and Fiona Petchey for advice on radiocarbon-dating data reporting, Xiao Fu, John D. Jansen, and David Price for contributions to the SahulSed Fluvial OSL and TL sub-collections, Richard Gillespie for contributions to SahulArch, and Corey Bradshaw and Frédérik Saltré for sharing information and data regarding FosSahul. We also thank Alan Young and Wenny Hidayat from Kasna for their work on migrating OCTOPUS to Google Cloud Platform. HM thanks Jessica Blois, Jack





Williams, Simon Goring, and Eric Grimm for insightful mentoring. Finally, we thank Richard G. Roberts and Julie Matarczyk for discussions and their support of the OCTOPUS project. We acknowledge the Traditional Custodians of the lands on which we have worked, and their continued spiritual and cultural connection to Country.



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
