# Peer review of "OCTOPUS Database v.2"

_Earth System Science Data, 2022_

## Author Response (AR1)

28 June 2022

Dr. Alexandru T. Codilean
Senior Lecturer in Earth Surface Processes

**To: Editors of Earth System Science Data**

Dear Dr. Kirsten Elger

Thank you once again for agreeing to handle our manuscript.

We have now made revisions to the manuscript following comments from the two reviewers. Changes required were mostly of a cosmetic nature. In addition to these, at the prompting of reviewer #1, we have also updated four of our figures to make these more accessible for those with a visual impairment or for those printing the manuscript in black and white.

Our responses to every comment were already posted on the EESD interactive discussion page, however, we copy these here also.

Sincerely,

Alexandru T. Codilean

**School of Earth, Atmospheric and Life Sciences**
**Faculty of Science, Medicine and Health**

UNIVERSITY OF WOLLONGONG, NSW 2522 AUSTRALIA

P (+61) 2 4221 3013  F (+61) 2 4221 4135

seals-admin@uow.edu.au  smah.uow.edu.au  CRICOS PROVIDER No. 00102E

**OCTOPUS Database v.2 – Codilean et al. 2022 ESSD**

**Responses to Reviewer #1**

| Line | Text | Reviewer comment | Author response |
|---|---|---|---|
| 2 | download | could say a range of data main topic X with additional types etc. | This is difficult to rephrase as suggested by the reviewer given the range of different dataset types. Although might be a bit awkward, we do not think that this sentence lacks clarity. |
| 4 | FAIR | a tough one is it really FAIR? I can see that you have provisioned I and R through WFS and *R* linkage, however do machines connecting to Octopus 'understand' the data | Machines connecting to OCTOPUS must understand the data given that they connect via WFS. For example when connecting to the database using QGIS, the only information required from the user is the URL, and everything else is automatically retrieved by QGIS via the GetCapabilities command. The latter returns a list of standard XML instructions that are machine readable. Further we have DOIs minted for every collection and so we are confident that we tick the FAIR boxes to the extent where we can claim that our database follows FAIR 'data principles'. |
| 6 | download | You've already mentioned these in line 1-2 (double up - unnessary?) | OK. Replace with 'accessed' |
| 7 | respectively | style makes it difficult to understand which data collection has what stored, maybe a 1. - 2. - 3. etc with all things in those collections is easier for readers | We do not find this sentence difficult to understand. Both CRN and ExpAge collections include both 10Be ans 26Al data. The CRN collection is about denudation rates and ExpAge about exposure ages in glacial landforms. |
| 9 | the former also | replace with "CRN Denudation" | OK. Replaced as suggested by reviewer |
| 14 | data | add "to all set" | OK. Replaced as suggested by reviewer |
| 32 | Like | new paragraph | OK. Replaced as suggested by reviewer |
| 38-44 | Furthermore | This is an important part and difficult to follow, you introduce the reasons for why you have to collate/curate/clean data for Octopus (in one sentence) | OK. Replaced as suggested by reviewer |
| 44-47 | The above | This can also be rephrased I suppose, to something like "To ensure longevity and safeguard often irreplaceable legacy data of CRN, ..., that suffers the above limitations curation in a single system is worth the effort..." | We decided to keep this sentence as is given that what the reviewer suggests would change the intended meaning. |
| 51 | to | remove | Our formulation is correct as the database is "served to" the user community. |
| 61 | seal | sea? | OK. Replaced as suggested by reviewer |
| 62 | direct and indirect | I don't quite understand the meaning here? A bit convoluted. | A fossil may be dated directly (e.g., via radiocarbon dating of the fossil itself) or indirectly by dating the sedimentary deposit surrouding the former. To avoid potential confusion from readers, we removed this from the text. |
| 63-64 | Supporting | Sentence occurs almost the same in abstract ok? | We think this should be OK as we are re-iterating an important aspect of the database. |
| 66 | recalculated and harmonised | Fantastic work to have done that! | No action required here. |

| | | | |
|---|---|---|---|
| 70 | modular | can add reference to paper XY 2018 | We are not sure what paper the reviewer is referring to here or whether there is anything that should be cited here. GCP is by definition modular and what we are doing with OCTOPUS v.2 is breaking up a monolithic design and modularising it as much as possible. |
| 84 | that is | add 'now' | OK. Replaced as suggested by reviewer |
| 87 | ocg | is 'OGC' ? | OK. Replaced as suggested by reviewer |
| 93-98 | More recent… volume | Note clear if this is current or planned for the future. | These are modifications planned / desired for the future |
| fig 3 | | transferring some explanations from the text would help the reader here if they just want to scan through and get an idea of how the DB works | Transfering explanation from the text would make the figure caption very large and probably best if this figure is looked at in conjunction with reading the main text -- as it is a complex figure. |
| 122-125 | Finally, … / reproduced | Cool, any quick way for a user to assess the quality of the data here? | All of this data is included in the attribute table that users download along with the geospatial files. |
| 130 | Shell fragments… | Sentence should be inserted after 'and' in line 129 to make it more comprehensive. Have you thought about IGSN here? Regarding the assigning of Octopus unique identifiers? | Yes and in fact the database includes an IGSN placeholder field that is currently unpopulated. The main 'difficulty' we are facing is that the database compiles published data and so we are reluctant to mint IGSN identifiers for samples that other people/groups have collected/own. In our view this would potentially generate a lot of confusion and might also alienate some members of the scientific community whose data we include in OCTOPUS. |
| 139-140 | When… (n=19) | Have these been published as say vocabularies that can be used to become interoperable think of Research Vocabularies Australia as a vocab service | No but this is an excellent suggestion that we will consider in the future. |
| 146-150 | | are they recalculated and entered/stored again or can they be recalculated multiple times (on the fly?)? | The answer to this point is both Yes and No. The recalculation of denudation rates is not trivial and requires to be done offline. It is possible to do approximate calculations on the fly by passing data to existing web-based calculators (such as this one: https://hess.ess.washington.edu) via a third party applications. However, as part of the OCTOPUS database we provide everything necessary for users to perform this recalculation. Users need to download and install a third party application and then run it with the files provided. The third party application -- CAIRN -- is open source and documented in the literature. |
| 152-153 | | Does the db show any of these parameters? | Yes, all of this information in included in the database and is downloaded as part of the attribute table. |
| 161-162 | Users…reproducible | Where does a user publish these? Is this part of Octopus v2? | We acknowledge that our choice of words may be confusing. Essentially as long as input parameters for CAIRN and the DEM used are provided by users, denudation rates will be reproducible. |
| 167 | | Have you thought about collections organisation when multiple collections come online (say organisation of menus on the collection activation/selection). | Yes -- we believe this is achieved by the current version of the web interface. |

| | | | |
|---|---|---|---|
| 177 | respective recalculated exposure | does tit include how that was done (paper describes this but apparent in db?)? | Yes -- old and new ages are provided in the database and what is what is made clear to users. |
| 179 | | do they have a clear timestamp (of recalculation)? | Yes, the ExpAge database has clear time stamps. Same is true for FosSahul. |
| 186 | The two… | talking about which collections? | OK. Replaced as suggested by reviewer |
| 186-191 | The two… | confusing | OK. Replaced as suggested by reviewer |
| 198 | is similare to | how does octopus 2 differ from octopus 1 | We believe this question refers to how CRN Denudation v.1 is different from CRN Denudation v.2? What we are saying on line 198 is that although with CRN Denudation v.2 we are adding another 1000+ data points, their geographic distribution is similar to what we had before. In other words, unfortunately CRN Denudation v.2 does not improve the under-representation of certain geographic areas. However, this is beyond our control as we are limited by what data is available in the published literature. |
| 206 | & | replace with 'and'? | OSL & TL was the name of the data collection and so we need to keep the "&" here. |
| 206-218 | | Sahul Sed is the same and better than OSL and TL Australia, except for… that can still be found here… | We do not think changing the text here is necessary. |
| fig 4 | legend | match the format of the legend to the format of the individual plots (as in all legend bars horizontal in line with each other and their descriptions underneath) | We are not sure we can see a problem with how the legend is formatted at the moment. We could flip the bars to be vertical to resemble better the box plots, but placing the description underneath would make them look awkward given that the explanation text is much longer than the bars in some cases. |
| table 2 | s### | suppose this means the sample? What part of the sample? Table caption can explain this. | s### where ### is a 3 digit number refers to the publication from where data is compiled from. We now explain in table to avoid confusion. |
| 241 | Fig 5D | I see it is one figure due to need for comparison between TL and OSL (initially I thought this fig would look better if OSL and TL were separated but no need) legend can again be horizontally formatted with text OSL / TL underneath | We will make changes to this figure in line with reviewer suggestions. |
| Page 13 | | This is fantastic! How does it compare to Octopus 1.0 and how will IT (guessing I meant SahulArch) be updated / maintained or will this be a time stamped version with new version releases to happen in future | In Table 1 we provide an estimate of how complete each data collection is. For example CRN Denudation Australia is >99% complete, i.e., we are confident that we have captured all existing published data. SahulArch Radiocarbon, on the other hand is about 50% complete, i.e., of the data we know exists, we only managed to compile 50% by the time we released OCTOPUS v.2. As of today we are closer to 100% and a new version of SahulArch will be release later this year. To asnwer the reviewer, yes, SahulArch is a time stamped version (see DOI in Table 1) and new version will have their own time stamps and DOIs. In fact, every database tuple has two timestamps: one for generation (CREATED_AT) and one for recording changes (UPDATED_AT). More details about the relational database are found in our SchemSpy generated output (cited in manuscript and also found here: https://zenodo.org/record/5874855#.YrljpS0RrAw). |

| | | | |
|---|---|---|---|
| fig 5 | | Bunch of design feedback attached image | We will make changes to this figure in line with reviewer suggestions. |
| 259 | observations | is this the 50% completion number or the total? | This is the 50% number. Please have a look at Table 1 for more info. |
| 263 | the IQR | be consistent here with other mentioning of IQR and median in document e.g.: 'IQR "low to high" and median being 20 with ~7 observations' | OK. Replaced as suggested by reviewer |
| 264 | continent of Sahul | it be great to explain Sahul a little earlier in the paper for people like myself :) | OK. This is a good idea. |
| 275 | | any explanation here? You've discussed possible biases etc. in previous collection descriptions. | No explanation necessary here. These differences are driven by the differences between the three dating techniques. |
| sect. 5.4 | | See image for feedback attached (reorganisation of first couple of sentences. | OK. We will consider the feedback from the reviewer to clarify the text. |
| 292 | recalculated | (... how) CAIRN? | This is a good point and we have clarified in the text. |
| 293 | 16009 | is that also 50%? | The 50% only pertains to SahulArch -- see Table 1. |
| 294 | metasites | I forget what this means again, suppose it is the location (lat/long) | Metasites are a conceptual construct defined as "agglomeration of sites sharing common properties" -- as explained using the Glen Lossie example in Figure 3. |
| 308 | fig 7 | insert weblink! | OK. Replaced as suggested by reviewer |
| 308 | GeoServer | how to access / say if I were to connect via QGIS? Is that possible (explained later on I think) | Yes, this is explained later in the text. |
| 314-320 | | Excellent! | No action required here. |
| fig 7 | | Include explanation of A-J in figure caption; include weblink in figure caption (perhaps include 1. 2. 3. 4. 'titles' in caption) | We will include weblink but we are reluctant to expand the figure caption with explanations of A-J. The latter is explained in the text and including in caption would make this too long. Also, like Figure 3, this figure is best looked at in conjunction with reading the main text. |
| 343 | of the data | include (fig 7, xy) mention | OK. Replaced as suggested by reviewer |
| 343 | be used for | include 'OCTOPUS' | OK. Replaced as suggested by reviewer |
| 344 | further funding to support octopus | replace with 'future OCTOPUS support funding' | OK. Replaced as suggested by reviewer |
| 348 | filter. | are users aware of this on the web-application? | We have prepared a user guide which is deposited on Zenodo. However, there is currently nothing on the web interface warning users about this. However, our aim is for the current paper to become the main 'help resource' that users access. We will include a reference to the manual on Zenodo and add it to the list of assets associated with this paper. |
| 355 | A valid ... is pressed | CLEVER! | No action required here. |
| 360 | expansion of | replace with 'expanding' | OK. Replaced as suggested by reviewer |
| fig 8 | | nice overviews (could use a different line for New Data as in a dash-dot-dash for example) | Is the reviewer refering to the 1:1 line? If yes, we are not sure we see why this would be useful. However, in line with comments for Figure 5, we will modify this (and other) figure(s) to make them more accessible when printed in b&w. |
| 412 | is capture | excellent, how can users download / access that? | This all comes with the data as part of the attribute table |
| 430-431 | should... convention. | What about lists of terms used? Are these available somewhere? Where? Think of Research Vocabularies Australia | No but this is an excellent suggestion that we will consider in the future. |

| 436 | being added | oef, no updates possible? | Again, the answer to this question is complicated. Updates are not done immediately and continuosly. Rather, the database is updated with periodic release of new versions. We provide a lengthier response to this as part of our response to Reviewer #2. |
|-----|-------------|---------------------------|--------------------------------------------------------------------------------------------------------------------------------------------------------------------------------------------------------------------------------------------------------------------|
| 449 | resource…archaeology | what about Global collection? | OCTOPUS is already an important resource to the global community -- as illustrated with Figure 1. |
| 454 | possible. | could you include a call to action of sorts to motivate the community to contribute or use your templates list of terms etc.? | We plan to do this via other avenues rather than via this manuscript. See also our response to Reviewer #2. |

**OCTOPUS Database v.2 – Codilean et al. 2022 ESSD**

**Responses to Reviewer #2**

RC: Reviewer comment
AR: Autor response

| | |
|---|---|
| RC1 | The article of Codilean et al. presents the updated OCTOPUS database, which is a resource for cosmogenic, OSL and radiocarbon ages including both their metadata and geological/archaeological context. I applaud the authors for this initiative which is an important step forward in ensuring that the metadata necessary for age evaluation/or recalculation is appropriately archived, and which will also facilitate the integration of geochronological data beyond the individual study for which the data were originally compiled. |
| AR1 | We thank the reviewer for the positive feedback and agree with them that OCTOPUS is an important step in ensuring that cosmogenic, luminescence, and radiocarbon data remain a reusable resource after publication. Over the last four years, OCTOPUS has become an important resource to the global geomorphology community, and we hope that with the expansion of SahulSed and the addition of SahulArch, OCTOPUS v.2 will become an equally important resource for the Quaternary and archaeology research communities. |
| RC2 | I cannot comment on the technical details of the database or its structural organisation as this is beyond my expertise, however I did not see any obvious issues. I found the distinction of the SahulArch database as archaeological ages a slightly odd distinction relative to the sedimentary ages – archaeological sites are also sedimentary archives and interpretation of any age is dependent on understanding its depositional setting. |
| AR2 | Indeed, archaeological sites are also sedimentary archives and interpretation of OSL/TL and radiocarbon ages is dependent on understanding of the depositional context. The OCTOPUS relational database reflects the above in its design. For example all OSL, TL, and radiocarbon 'observations' (i.e., ages in OCTOPUS parlance) are stored in method-specific data tables that share the same organisation. Thus, at the database level, there is no separation between OSL, TL, or radiocarbon ages based on their depositional settings. The depositional setting, however, means that an archaeological, fluvial, aeolian, or lacustrine OSL sample, for example, will have different contextual information that needs to be collected and stored. The latter are stored in separate tables that are then linked to the tables that store the ages.

Separating SahulSed from SahulArch and further splitting the two collections into different sub-collections (e.g., SahulSed in six sub-collections and SahulArch in three) has several advantages, including: |

| | |
|---|---|
| | (1) Smaller datasets mean that data will load faster when accessed via the OCTOPUS webpage – for example see the difference in loading speeds (and user experience, more generally) between any of the SahulSed sub-collections and the two largest collections, namely, FosSahul and ExpAge

(2) Separating into thematic sub-collections means that data tables do not need to contain redundant information (e.g., a SahulSed Fluvial data table does not need to contain fields specific to archaeological sites)

(3) Given their culturally sensitive nature, archaeological site coordinates have been randomly obfuscated within a 25km radius. Serving SahulArch as a separate collection means that the same obfuscation does not need to be applied to SahulSed data, as it would have been the case were the two one single collection.

Notwithstanding the above, the semantic data model that we have designed for the OCTOPUS relational database means that although SahulArch and SahulSed are served as two separate collections, they can 'talk' to each other and nothing is lost by serving them separate. |
| RC3 | Also, it is a little odd that the authors state that they make no judgement on the quality of ages, but the FosSahul database includes quality rated data. I agree that it makes no sense to remove this information from the FosSahul data, but are there minimum reporting standards imposed by OCTOPUS for the inclusion of data from grey literature? This should be explicitly stated. |
| AR3 | Our manuscript explicitly sates our approach to data inclusion and our editorial thresholds in Section 7.

Lines 382 – 285: "*As with the previous version of OCTOPUS, our aim was to compile and incorporate all data — both published and unpublished — that is publicly available. It is not our role to decide on the quality of the data that are already published and so we make no editorial decisions on what data to include or exclude. Further, we designed the database in a way that it captures sufficient auxiliary information for users to be able to make informed judgements regarding data quality, themselves.*"

Lines 408 – 412: "*Unlike CRN-based exposure ages and denudation rates, the OSL, TL, and radiocarbon data compiled as part of SahulSed and SahulArch do not require recalculation. For this reason, the two data collections are verbatim reproductions of what was present in the various sources that the data was compiled from. We do not quality-rate the published OSL, TL, or radiocarbon ages, however, to enable end users to undertake such quality rating if desired, we designed the database so that a wide range of method related data is captured.*"

To summarise, our philosophy is to include all ages available and present all the information possible for the user to be able to decide on suitability / quality later. |

| | |
|---|---|
| | This philosophy also extends to our partner data collections, such as FosSahul -- in this case we retain all information found in the published version, including quality rating of ages. |
| RC4 | At present the compiled datasets are somewhat limited geographically, as acknowledged by the authors. There are other databases of geochronological data, that could be incorporated into Octopus and I am curious as to why the authors have not yet included them (already too much to include? I note that some of the other datasets are only partially incorporated). For example, the INQUA Dunes Atlas (Lancaster et al., 2016, Quaternary International) could be incorporated, increasing the global distribution of data. Individual laboratories also have some publicly accessible databases eg. https://www.lumid.nl/ from the Wageningen University luminescence laboratory. |
| AR4 | Although OCTOPUS already includes global coverage of cosmogenic data, our ultimate hope is that the same will apply to OSL, TL, and radiocarbon data in the future. However, there are several issues to consider:

(1) While each database is valuable, this value diminishes considerably once the data are no longer supported / updated. We recognise our limitations and so our strategy has been to produce data collections that we are able to maintain. Given the relatively small number of data points, we have been able to support a global compilation of cosmogenic radionuclide data and as shown in Table 1, we managed to capture 75% of available data in this OCTOPUS release. OSL/TL and radiocarbon data, on the other hand are a different order of magnitude in terms of number of data points. Further, the lack of reporting standards and the diversity of the OSL/TL and radiocarbon user communities means that these data will also be messier – and so more effort to compile. For example with SahulArch, we include roughly the same number of data points as we have in the global cosmogenic radionuclide collection, but we only capture 50% of the available data. Thus, instead of compiling data at the global scale and producing a database that is a one-off (such as the INQUA Dunes Atlas), our focus is on collections that we can maintain over the longer-term.

(2) The above does not mean that we shut out completely other databases. For example, we could add the INQUA Dunes Atlas or any other existing database as a partner collection. We would of course need the collaboration of those who produced these databases, and there would need to be a community desire for these things to happen given the practical and logistical challenges (i.e., the need for funding).

To summarise, we are keen and open to expand OCTOPUS, but we need to do this in a way that is sustainable. This is the reason why we have limited ourselves geographically in some instances. |

| RC5 | I was disappointed that although there is a call for contributions from other groups to the database, that it was somewhat hidden late in the manuscript and that the form specifying the metadata required is only available from the authors – why not make it available for download from the OCTOPUS website with the link in this manuscript? Whilst reporting standards have not yet been agreed (by the luminescence dating community at least), OCTOPUS has effectively imposed some standards from the original template design, or could these templates be modified? |
|-----|-----|
| AR5 | As with RC4, this is a complex issue.

As we state in the manuscript, users can download data from OCTOPUS and use that data as a template for any user contributions. However, there is always an advantage in contacting us first.

Given that data are stored in a relational database, adding data is a multi-step process, undertaken by multiple people. First, the data is collected from publications in a spreadsheet and is then processed so that the correct metasite, site, sample, and observation identifiers are added, and the data is split into the various tables making up the relational database. The former (data collection) does not require prior knowledge of the relational database structure. The latter (assigning of identifiers and splitting into constituent tables), on the other hand, requires intimate knowledge of the database and thus requires substantial training.

To facilitate with the initial data collection, we have created templates for a data collection software called E4 (https://www.oldstoneage.com/osa/tech/e4/) and having potential contributors contact us beforehand means that we can provide them with these resources and the training needed and so make the data compilation process easier for both sides.

Rather than rely on, or encourage, random user contributions, our goal is to democratise OCTOPUS data collection by following the example of the Neotoma Paleoecology Database (https://www.neotomadb.org/data/category/contribution) and train a community of 'data stewards' who can then focus on specific themes or sub-collections. The above will ultimately mean that we can potentially maintain global compilations without compromising on the breath of metadata that we include. |